

# Long-chain diols in rivers: distribution and potential biological sources

Julie Lattaud[1*], Frédérique Kirkels[2], Francien Peterse[2], Chantal V. Freymond[3], Timothy I. Eglinton[3], Jens Hefter[4], Gesine Mollenhauer[4], Sergio Balzano[1], Laura Villanueva[1], Marcel T.J. van der Meer[1], Ellen C. Hopmans[1], Jaap S. Sinninghe Damsté[1,2] and Stefan Schouten[1,2]

[1]NIOZ Royal Netherlands Institute for Sea Research, Department of Marine Microbiology and Biogeochemistry, and Utrecht University, The Netherlands.

[2]Utrecht University, Department of Earth Sciences, Faculty of Geosciences, Heidelberglaan 2, 3584 CS Utrecht, The Netherlands.

[3]Geological Institute, ETH Zürich, Sonneggstrasse 5, 8092 Zürich, Switzerland.

[4]Alfred Wegener Institute, Department of Geosciences, Marine Geochemistry, Am Handelshafen 12, Bremerhaven, Germany.

*Corresponding author: Julie.lattaud@nioz.nl

To be submitted to: Biogeosciences

**Abstract.** Long chain diols (LCDs) occur widespread in marine environments and also in lakes and rivers. Transport of LCDs from rivers may impact the distribution of LCDs in coastal environments, however relatively little is known about the distribution and biological sources of LCDs in river systems. In this study, we investigated the distribution of LCDs in suspended particulate matter (SPM) of three river systems (Godavari, Danube, and Rhine) in relation with season, precipitation, temperature, and source catchments. The dominant long-chain diol is the $C_{32}$ 1,15-diol followed by the $C_{30}$ 1,15-diol in all studied river systems. In regions influenced by marine waters, such as delta systems, the fractional abundance of the $C_{30}$ 1,15-diol is substantially higher than in the river itself, suggesting different LCD producers in marine and freshwater environments. A change in the LCD distribution along the downstream transects of the rivers studied was not observed. However, an effect of river flow is observed, i.e. the concentration of the $C_{32}$ 1,15-diol is higher in stagnant waters, such as reservoirs and during seasons with river low stands. A seasonal change in the LCD distribution was observed in the Rhine, likely due to a change in the producers. Eukaryotic diversity analysis by 18S rRNA gene sequencing of SPM from the Rhine showed extremely low abundances of sequences (i.e. <0.32% of total reads) related to known algal LCD producers. Furthermore, incubation of the river water with $^{13}C$-labelled bicarbonate did not result in $^{13}C$ incorporation into LCDs. This indicates that the LCDs present are mainly of fossil origin in the fast flowing part of the Rhine. Overall, our results suggest that the LCD-producers in rivers predominantly reside in lakes or side ponds that are part of the river system.



## 1 Introduction

Long-chain diols (LCDs) occur widespread in marine environments and have been shown to mainly consist of $C_{28}$ and $C_{30}$ 1,13-diols, $C_{30}$ and $C_{32}$-1,15-diols (Versteegh et al., 1997, 2000; Gogou and Stephanou 2004; Smith et al., 2013; Schmidt et

al., 2010; Rampen et al., 2012, 2014a; Balzano et al., 2018) and $C_{28}$ and $C_{30}$ 1,14-diols (Sinninghe Damsté et al., 2003; Rampen et al., 2011, 2014a). Culture studies showed that eustigmatophyte algae produce 1,13 and 1,15-diols (Volkman et al., 1999; Rampen et al., 2007, 2014b), but with a distribution different than encountered in marine sediments. Furthermore, eustigmatophyte algae are rarely reported in marine environments, indicating that the cultivated marine eustigmatophytes are likely not the main producers of 1,13 and 1,15-diols in marine environments (Volkman et al., 1999). Culture studies of

*Proboscia* diatoms show that they produce mainly 1,14-diols and minor amounts of 1,13-diols (Sinninghe Damsté et al., 2003; Rampen et al., 2007), while the marine dictyochophycean alga *Apedinella radians* (Rampen et al., 2011) also produces 1,14-diols. *Proboscia* diatoms are mainly present in upwelling areas and are likely the main source of 1,14-diols in nutrient-rich marine environments (Rampen et al., 2008; Willmott et al., 2010; Gal et al., 2018).

LCDs also occur in freshwater environments, i.e. in lakes (Shimokawara et al., 2010; Castañeda et al., 2011; Zhang et al.,

2011; Romero-Viana et al., 2012; Rampen et al., 2014b; Villanueva et al., 2014) and in rivers (De Bar et al., 2016; Lattaud et al., 2017a). Shimokawara et al. (2010) showed that the LCD distribution in Lake Baikal was similar to that of cultivated freshwater eustigmatophytes, indicating that they may be a source of LCDs in freshwater environments. In addition, Villanueva et al. (2014) observed a correlation between the LCD concentration in the water column of Lake Challa and the abundance of 18S rRNA gene copies of uncultivated eustigmatophytes. Rampen et al. (2014b) studied the LCD distribution of several

freshwater eustigmatophyte cultures, showing that for the Goniochloridaceae and Monodopsidaceae families the main LCD is the $C_{32}$ 1,15-diol, while the LCDs of members of the Eustigmataceae family are dominated by a mix of $C_{28}$ 1,13-, $C_{30}$ 1,15- and $C_{32}$ 1,15-diols. Interestingly, an increase in the temperature at which these algae were cultivated resulted in an increase of the fractional abundance of the $C_{32}$ 1,15-diol (Rampen et al., 2014b). Apart from lakes, LCDs have recently been reported to occur also in rivers (De Bar et al., 2016; Lattaud et al., 2017a) with the $C_{32}$ 1,15-diol and, to a lesser extent, the $C_{30}$ 1,15-diol

as the most abundant LCDs. In contrast to marine and lake systems, however, the spatial occurrence and sources of LCDs in river systems have not been studied in detail.

In this study we investigated three river systems, i.e. the Rhine, Godavari, and Danube rivers, to constrain the impact of river characteristics on the distribution of LCDs. Furthermore, we analyzed the algal community composition using 18S rRNA gene



sequencing and quantitative PCR (qPCR) analysis and performed labelling studies in the Rhine to constrain the biological sources of LCDs.

## 2. Material and methods

### 2.1. Material

**2.1.1. Godavari**

The Godavari is the largest river of India not draining the Himalayas with a catchment area of $310 \times 10^3$ km$^2$ (Balakrishna and Probst, 2005) and a length (of the main river) of 1465 km (Ramasubramanian et al., 2006). The principal tributaries of the river are the Pranhita, Waiganga and Wardha forming a subcatchment (called Pranhita) in the North and the Indravati and Sabari draining the Eastern Ghats (called Indravati). In addition to these subcatchments, the main stem of the Godavari River can be

divided into the Upper Godavari (from the source to Sironcha), the Middle Godavari (from Sironcha to the Dowleswaram reservoir) and the lower Godavari (downstream of the reservoir). The climate over the basin is semi-arid to monsoonal (10 to 45°C, Biksham and Subramanian, 1988) and rainfall increases along a west-east gradient. Maximal rainfall is experienced during the Southwest monsoon over India (annual rainfall is 1185 mm with 84% falling during monsoonal months, June to September, Biksham and Subramanian, 1988; Pradhan et al., 2014). Ca. 98 % of the total suspended sediment load of the

Godavari River is transported during the monsoon period (Rao et al., 2015).

A total of 62 (20 in dry and 42 in wet season) SPM and 65 (34 in dry and 30 in wet season) riverbed sediments were collected as described by Usman et al. (2018) during the dry (February/March) and the wet (July/August) seasons in 2015 (Fig. 1a). For SPM sampling, 5-50 L of river water was filtered over a pre-ashed glass fiber filter (Whatman GF-F, 0.7 µm, 142 mm diameter). Riverbed sediments were collected from the middle of the stream from bridges, using a van Veen grab sampler.

### 2.1.2. Danube and Black Sea

The Danube is the second largest river of Europe with a catchment area of $800 \times 10^3$ km² and a length of 2850 km (Freymond et al., 2017). Its catchment can be divided into four sub-regions: the upper Danube (from the source to the Gate of Devin), the Middle Danube (from the Gate of Devin to the Iron Gates reservoir) and the lower Danube (downstream of the Iron Gates

reservoir to the delta) and the delta. The Danube flows directly into the Black Sea through three main branches of its delta: Chilia, Sulina, and Sf. Gheorghe. In the Black Sea, south of the Crimea peninsula, surface water flows westward onto the northwestern shelf. On the shelf, the surface current turns southward along the coast (Tolmazin, 1985). The climate in the Danube catchment is diverse, with an oceanic influence on the western part of the upper basin, a Mediterranean influence in the south and central part of the middle basin and a continental climate influence in the other parts. The annual precipitation

varies from 2000 mm.y$^{-1}$ in the mountain area to 500 mm.y$^{-1}$ in the plains (Rimbu et al., 2002).

46 riverbed sediments (main river and tributaries, Fig. 2a) were collected as described by Freymond et al. (2018) in spring 2013 and 2014. The riverbed sediments were wet sieved with milliQ water over a 63 µm sieve on a shaking table and the fine fraction (<63 µm) was studied. In total 14 surface sediments from the Black Sea were included in this study. Of these, 10 surface sediments (Fig. 2a) were obtained as described by Kusch et al. (2010, 2016; called the Meteor/Poseidon surface

sediments), one was collected in 2016 by the R/V Pelagia during the cruise 64PE408, and three were collected in 2017 with the R/V Pelagia during the cruise 64PE418 (collectively called the Pelagia surface sediments).

### 2.1.3. Rhine

The Rhine is the third largest river of Europe with a catchment area of $185 \times 10^3$ km² and a length of 1320 km (Hoffmann et al.,

2007). It can be divided into three hydrological areas: the upper Rhine (the Alpine region), the Middle Rhine (German and French Rhine) and the lower Rhine (the delta region). The upper Rhine receives up to 2000 mm of precipitation per year, the middle Rhine is characterized by a temperate oceanic climate, with annual rainfall ranging from 570 to 1100 mm and the lower Rhine receives an average of 800 mm of rain per year and has a temperate oceanic climate influenced by the North Sea and the Atlantic Ocean (Pfister et al., 2004).

Five sites along the middle Rhine were sampled (Fig. 3a): Karlsruhe (station of the Landesanstalt für Umwelt, Messungen und Naturschutz Baden-Württemberg), Mainz (station of the Landesamt für Umwelt, Wasserwirtschaft und Gewerbeaufsicht), Koblenz (station of Landesamt für Umwelt Rheinland-Pfalz), Cologne (Ecological Rhine-Station of the University of Cologne) and Kleve-Bimmen (International monitoring station). 62 L of river water was collected manually with a bucket at the side of the river at each station in March and September 2016. Of this 20 L was used for LCD analysis, 1 L for DNA analysis and 500

mL for chlorophyll analysis and 40 L was used for stable carbon isotope incubations.

### 2.1.4. Isotope incubation

For the incubation experiments, two 20 L Nalgene bottles were filled with Rhine water and incubated in presence of light at ambient air temperature during 52 h (day/night cycle) with 100 mg $^{13}$C-labelled bicarbonate (Cambridge Isotope Laboratories,

Inc., USA). The bottles were shaken at the start of the incubation, then once a day to avoid particles sinking to the bottom of the bottle. The bottles were not sealed so gas exchange with the atmosphere was possible. The water was filtered using pre-ashed glass fiber filters (Whatman GF-F, 0.7 µm, 142 mm diameter) using a peristaltic pump (WTS, McLane Labs, Falmouth, MA). All samples were kept frozen at -20°C.

## 2.2. Methods

### 2.2.1. Lipid extraction



Filters of the Rhine and the incubation experiment were base hydrolyzed with 12 mL of 1 N KOH in methanol (MeOH) solution by refluxing for 1 h. Afterwards the pH was adjusted to 4 with 2 N HCl : MeOH (1 : 1, v:v) and the extract was transferred into a separatory funnel. The residues were further extracted once with MeOH : $H_2O$ (1 : 1, v/v), twice with MeOH, and three times with dichloromethane (DCM). The extracts were combined in the separatory funnel and bidistilled water (6 mL) was added. The combined solutions were mixed, shaken and separated into a MeOH : $H_2O$ and a DCM phase, after which the DCM phase was removed and collected into a centrifuge tube. The MeOH : $H_2O$ layer was re-extracted twice with 3 mL DCM. The pooled DCM layers were dried over a $Na_2SO_4$ column and the DCM was evaporated under a stream of nitrogen. The extract was then acid hydrolyzed with 2 mL of 1.5 N HCl in MeOH solution under reflux for 2 h. The pH was adjusted to 4 by adding 2 N KOH : MeOH. 2 mL of DCM and 2 mL of bidistilled water were added to the hydrolyzed extract, mixed and shaken and, after phase separation, the DCM layer was transferred into another centrifuge tube. The remaining aqueous layer was washed twice with 2 mL DCM. The combined DCM layers were dried over a $Na_2SO_4$ column and the DCM was evaporated under a stream of nitrogen.

The SPM filters and riverbed sediments from the Godavari were freeze-dried, and river sediments were homogenized by milling. Both filters and riverbed sediments were extracted as described by Usman et al. (2018). Briefly, extraction was performed (3x) using an Accelerated Solvent Extractor (ASE 350, Dionex, Thermo-Scientific, Sunnyvale, CA, USA) with 9 : 1 (v/v) DCM : MeOH at 100°C and $7.6 \times 10^6$ Pa. The extracts were dried under $N_2$ and an 80% aliquot was further processed for analysis. The samples from the Danube were extracted as described by Freymond et al. (2018) using microwave–extraction (MARS) with 9 : 1 DCM : MeOH (v/v, 25 min at 100°C). The Meteor/Poseidon Black Sea surface sediments were extracted three times ultrasonically with a 9 : 1 DCM : MeOH (v/v) solvent mixture after addition of 1.96 µg of $C_{22}$ 7,16-diol as internal standard. The four Pelagia Black Sea surface sediments were extracted using an ASE with a DCM : MeOH mixture 9 : 1 (v/v) and a pressure of $7.6 \times 10^6$ Pa at 100°C.

### 2.2.2. Separation of the lipid extract

To the total lipid extracts of the Rhine SPM, the incubation experiment and the Pelagia Black Sea surface sediments an internal standard was added ($C_{22}$ 5,17-diol). They were subsequently separated into 3 fractions on an $Al_2O_3$ (activated for 3 h at 150°C) column. The apolar fraction was eluted with 4 column volumes of 9 : 1 (v/v) hexane (hex) : DCM, the ketone fraction with 3 column volumes of 1 : 1 (v/v) hex : DCM and the polar fraction (containing the diols) with 3 column volumes of 1 : 1 (v/v) DCM : MeOH.

For all other samples, the $C_{46}$ glycerol trialkyl glycerol tetraether (GDGT, Huguet et al., 2006) was added as an internal standard.

The Godavari SPM from the wet season and Danube (Freymond et al., 2018) total extracts were saponified with KOH in MeOH (0.5 M, 2 h at 70°C). 5 mL of MilliQ water with NaCl was added and the neutral phase was back extracted with hexane and further separated into an apolar and polar fractions on a $SiO_2$ column with Hex : DCM (9 : 1, v/v) and DCM : MeOH (1 :





1, v/v), respectively. For the Godavari river sediments and SPM from the dry season, the total extracts were saponified with KOH in MeOH (0.5 M, 2 h at 70°C) and subsequently separated on a SiO$_2$ column with Hex : DCM (9 : 1, v/v) and DCM : MeOH (1 : 1, v/v), respectively. The Meteor/Poseidon surface sediments total extracts were saponified with KOH in MeOH at 80°C for 2 hours. The neutral fraction was recovered in hexane and separated into an apolar and a polar fraction by silica

gel column chromatography using DCM : hex (2 : 1, v:v) for the apolar and DCM : MeOH (1 : 1, v/v) for the polar fraction.

### 2.2.3. Diol analysis

The polar fractions were transferred into GC vials and silylated with N,O-Bis(trimethylsilyl)trifluoroacetamide (BSTFA) and pyridine (10 µL each) and heated at 60°C for 30 min, after which ethyl acetate was added. All diols except the Meteor/Poseidon

Black Sea surface sediments were analyzed by gas chromatography (Agilent 7990B GC) coupled to mass spectrometry (Agilent 5977A MSD) (GC-MS) equipped with a fused silica capillary column (Agilent CP Sil-5, length 25 m x diameter 320 µm x film thickness 0.12 µm). The temperature program for the oven was as follows: starting at 70°C, increased to 130°C at 20°C/min, increased to 320°C at 4°C/min, held at 320°C during 25 min. Flow was held constant at 2 mL/min. The MS source is held at 250°C and the quadrupole at 150°C. The electron impact ionization energy of the source was 70 eV. The diols of the

Meteor/Poseidon Black Sea surface sediment were analyzed by GC-MS using an Agilent 6850 GC coupled to an Agilent 5975C MSD equipped with a fused silica capillary column (Restek Rxi-1ms, length 30 m x diameter 250 µm x film thickness 0.25 µm). The temperature program for the oven was as follows: held at 60°C for 3 min, increased to 150°C at 20°C/min, increased to 320°C at 4°C/min, held at 320°C during 15 min. Flow was held constant at 1.2 mL/min. The MS source is held at 230°C and the quadrupole at 150°C. The electron impact ionization energy of the source was 70 eV.

The diols were identified and quantified via SIM (Single Ion Monitoring) of the m/z= 299.3 (C$_{28}$ 1,14-diol), 313.3 (C$_{28}$ 1,13-diol, C$_{30}$ 1,15-diol), 327.3 (C$_{30}$ 1,14-diol) and 341.3 (C$_{30}$ 1,13-diol, C$_{32}$ 1,15-diol) ions (Versteegh et al., 1997; Rampen et al., 2012).

Absolute concentrations were calculated using the C$_{22}$ 5,17-diol as internal standard for the Rhine SPM, Meteor/Poseidon and Pelagia surface sediments and the C$_{46}$GTGT as internal standard for the Godavari sediments and SPM and Danube SPM.


### 2.2.4. Isotope analysis of $^{13}$C incubation experiment

Before carbon isotope analysis, the LCDs in the polar fraction of the filter extract from the incubation experiment were isolated using semi-preparative normal phase HPLC. Prior to injection the polar fraction was dissolved in 750 µL hex : isopropanol (99 : 1, v/v) and filtered over a polytetrafluoroethylene (PTFE) filter (0.45 µm pore size). 3x 250 µL was injected on an 1260

infinity LC system (Hewlett Packard, Palo Alto, CA, USA) equipped with a thermostated autoinjector, column oven, and a Foxy R1 fraction collector (Teledyne Isco, Lincoln, NE, USA) as described in De Bar et al. (2016). Briefly, the different diol isomers were separated over a normal phase semi preparative Alltech Econosphere silica column (250 mm x 10 mm; 10 µm)



at room temperature. After 35 min at 14% A (hex :isopropanol, 9 : 1, v/v) and 86% B (hexane) the mobile phase was adjusted to 100 % A in 1 min. It was then held at 100% A between 35 and 55 min. Finally, the column was reconditioned with 14% A in hex at 3 mL/min. The fractions were collected from 15 to 40 min every 30 s and analyzed by GC-MS as described above. The LCDs of interest eluted between 22.5 and 27.5 min and were collected in 3 pools. Pool 1 from 22.5 to 24.5 min (containing

100% $C_{32}$ 1,15-diol, 95% $C_{30}$ 1,15-diol), pool 2 from 25 to 26 min (containing 83% $C_{28}$ 1,14-diol and 5% $C_{30}$ 1,15-diol) and pool 3 from 26.5 to 27.5 min (containing 100% $C_{28}$ 1,13-diol, 17 % $C_{28}$ 1,14-diol and 100% $C_{30}$ 1,13-diol). These pools were analyzed using gas chromatography–isotope ratio mass spectrometry (GC irMS, ThermoFinnigan Delta$^{PLUS}$ isotope ratio monitoring mass spectrometer coupled to an Agilent 6890 GC via a Combustion III interface). The gas chromatograph was equipped with a fused silica capillary column (25 m x 320 μm) coated with CP Sil-5 (film thickness = 0.12 μm) with helium

as carrier gas (2 mL/min). The LCDs were silylated as described above using BSTFA with a known $\delta^{13}$C value of -32.2 ± 0.5‰. Subsequently, the LCDs were injected splitless at an oven temperature of 70 °C (injector temperature was 250°C), then the oven was programmed to 130°C at 20°C/min, and then at 20°C/min to 320°C/min at which it was held for isothermal (10 min). The isotopic values were calculated by integrating the masses 44, 45, and 46 ion currents of the peaks produced by combustion of the chromatographically separated compounds and that of $CO_2$-peaks produced by the $CO_2$ reference gas with

a known $^{13}$C-content at the beginning and end of the analytical run. The samples were analyzed in triplicate.

### 2.2.5. Glycerol Dialkyl Glycerol Tetraether (GDGT) analysis

GDGTs were analyzed from the polar fractions of the Rhine and Pelagia Black Sea surface sediments. Prior to GDGT analysis an aliquot of the polar fractions was filtered through a 0.45 μm PTFE membrane filter using hex : isopropanol (99 : 1, v/v).

Analysis were performed using Agilent 1260 UHPLC coupled to a 6130 quadrupole MSD in selected ion monitoring mode following the method described by Hopmans et al. (2016). The Branched *versus* Isoprenoid Tetraether (BIT) index was calculated according to Hopmans et al. (2004). This proxy reflects soil and river input into marine environments but is also affected by in-situ marine production of brGDGT (De Jonge et al., 2014; Sinninghe Damsté et al., 2016).

### 25   2.2.6. 18S rRNA gene sequencing analysis

18S rRNA gene sequencing analysis was performed exclusively on DNA extracted from the Rhine water. To this end, 1/8 of the "DNA" filter was extracted using the PowerSoil kit (QIAGEN, Valencia, CA) following manufacturer's instructions. To amplify the eukaryotic V4 region of the 18S rRNA gene, we used the universal forward primer V4F (5'-CCA GCA SCY GCG GTA ATT CC-3', *S. cerevisiae* position 565-584) and a reverse primer V4R (5′-ACTTTCGTTCTTGAT(C/T)(A/G)A-3′, *S.

cerevisiae* position 964-981) from Stoeck et al. (2010). PCR reactions were performed on 5 replicates for each sample and each reaction included about 6 ng DNA template, 1.75 μL of each primer, 25 μL MasterMix phusion, 1.5 μL of DMSO and 19.25 μL deionised nuclease-free water for a total volume of 50 μL. Specifically, PCR consisted of an initial denaturation at



98 °C for 30 s, 11 x [98°C for 10 s, 53°C for 30 s, 72°C for 30 s]; 17 x [98°C for 10 s, 48°C for 30 s, 7°C for 30 s] as described in Logares et al. (2012). The PCR products were stained with SYBR® Safe (Life Technologies, the Netherlands) and visualised on a 1% agarose gel. Bands were excised with a sterile scalpel and purified with Qiaquick Gel Extraction Kit (QIAGEN, Valencia, CA) following the manufacturer's instructions. Equimolar concentrations of the barcoded PCR products were pooled

and sequenced on GS FLX Titanium platform (454 Life Sciences) by Macrogen Inc., South Korea.

To estimate the concentration of total 18S rRNA genes of the Rhine SPM we carried out quantitative PCR (qPCR) using the same primers and the same cycling conditions as described above. qPCR analysis was performed on a Biorad CFX96TM Real-Time System/C1000 Thermal cycler equipped with CFX ManagerTM Software. Each reaction contained 12.5 µL MasterMix phusion, 8.25 µL deionised nuclease-free water, 0.75 µL DMSO, 1 µL from each primer and 0.5 µL Sybr green and 1 µL of

DNA template. Reactions were performed on an iCycler iQTM 96-well plates (Bio-Rad). A mixture of V4 18S rRNA gene amplicons obtained as described above was used to prepare standard solutions. All qPCR reactions were performed in triplicate with standard curves from 640 to $6.4 \times 10^8$ V4 18S rRNA molecules per microliter. Specificity of the qPCR was verified with melting curve analyses (50°C to 95°C).

### 2.2.7. Bioinformatic analyses

Bioinformatic analyses of the sequencing results  were carried out using the bioinformatic pipeline Quantitative Insight Into Microbial Ecology (QIIME) (Caporaso et al. 2010 ). 121 232 raw sequencing reads were cleaned and demultiplexed, then chimeras and singletons were removed as described previously (Balzano et al. 2015 ) for a final dataset consisting of 58 480 good quality reads. Sequences were clustered into operational taxonomic units (OTUs) based on 97 % sequence identity. The

dataset was then normalized by multiplying the percentage of reads with the concentration of V4 copies measured by qPCR. Relationships between LCDs and microbial eukaryotes were inferred by Spearman correlation analyses using the QIIME script observation_metadata_correlation.py and p-values were corrected for false discovery rate (Benjamini and Hochberg 1995).

### 2.2.8. Chlorophyll analysis

Rhine River water filters for pigment analysis were extracted following Holm-Hansen et al. (1965) and Arar and Collins (1997). Briefly, 20 mL cold acetone was added to the filters and stored in the fridge overnight. Subsequently, they were sonicated for 2 min in an ice bath to avoid chlorophyll degradation and 10 mL was transferred into a centrifuge tube and centrifuged for 10 min at 4000 rpm. 3 mL of the extract was then transferred into the observatory cuvette. The chlorophyll measurement was realized using a Fluorescence Spectrophotometer (Hitachi f-2500) calibrated with two standards containing

50 µg/L and 100 µg/L of chlorophyll a in acetone. Samples were measured one time to obtain Rb (fluorescence before



acidification) and another time when 2 drops of a solution of 10% hydrochloric acid was added to obtain Ra (fluorescence after acidification). Chlorophyll concentration was then calculated as followed:

$$Chl - a = \frac{(Rb - Ra) \times \frac{A}{B} \times V_{extracted}}{V_{filtrated}} \tag{1}$$

Where Chl-a is the chlorophyll a concentration, A and B are constants obtained by measuring the standards ($A = \frac{Chl - a_{standard}}{Rb_{standard}}$

and $B = 1 - \frac{Ra_{standard}}{Rb_{standard}}$).

## 3. Results

### 3.1. LCDs in the Godavari

The most abundant LCD in the SPM collected during the dry season is the $C_{30}$ 1,15-diol (average $500\pm520$ ng.L$^{-1}$, n=18) followed by the $C_{32}$ 1,15-diol ($200\pm170$ ng.L$^{-1}$) (Fig. 1c). The $C_{30}$ 1,13 and $C_{30}$ 1,14-diols occur in substantially lower

concentrations ($30\pm30$ ng.L$^{-1}$ and $30\pm20$ ng.L$^{-1}$, respectively), whilst the $C_{28}$ 1,13 and $C_{28}$ 1,14-diols only occur in even lower concentrations ($4\pm5$ ng.L$^{-1}$ and $4\pm6$ ng.L$^{-1}$ respectively) (Fig. 1c). In the SPM collected during the wet season, the concentration of total LCDs is significantly higher than during the dry season (t-test, $p<0.05$). The $C_{32}$ 1,15-diol is the most abundant of the LCDs in the wet season SPM (Fig. 1c, $740\pm710$ ng.L$^{-1}$, n=41), followed by the $C_{30}$ 1,15-diol ($500\pm530$ ng.L$^{-1}$), with much lower concentrations of $C_{30}$ 1,13 and $C_{30}$ 1,14-diol ($20\pm10$ and $30\pm20$ ng.L$^{-1}$, respectively).

In the Godavari riverbed sediments, the $C_{32}$ 1,15-diol and $C_{30}$ 1,15-diol are the most abundant LCDs. The $C_{32}$ 1,15-diol is higher in abundance in the sediments collected during the dry season than in the wet season sediments ($330\pm370$ ng.g$^{-1}$ sed, n=30; $160\pm240$ ng.g$^{-1}$ sed, n=34 respectively; Fig. 1b) except for the wet season sediments from the Dowleswaram reservoir where maximal abundances are found ($500\pm80$ ng.g$^{-1}$ sed, n=2).

### 3.2. LCDs in the Danube and Black Sea

In the Danube sediments the main LCD is also the $C_{32}$ 1,15-diol ($3600\pm1300$ ng.g$^{-1}$ sed, n=51), followed by the $C_{30}$ 1,15-diol and the $C_{30}$ 1,13-diol ($1500\pm1000$ ng.g$^{-1}$ sed and $500\pm100$ ng.g$^{-1}$ sed, respectively). Furthermore, sediment from the Iron Gates reservoir shows the highest concentration (t-test, $p <0.001$) of $C_{32}$ 1,15-diol ($5400$ ng.g$^{-1}$ sed, n=1, Fig 2b) in comparison with any of the other parts of the Danube River system ($1500\pm3200$ ng.g$^{-1}$ sed, n=22 in the Upper Danube; $3800\pm2400$ ng.g$^{-1}$ sed,

n=14 in the Middle Danube; $4400\pm3400$ ng.g$^{-1}$ sed, n=10 in the Lower Danube; and $3000\pm1400$ ng.g$^{-1}$ sed, n=4 in the delta). For the Black Sea sediments (n=14) the $C_{30}$ 1,15-diol is the main LCD, with the two sediments from sites located closest to the river mouth (P128 and P177, Fig. 2a) having a lower fractional abundance of this diol (0.57) than all other sediments (0.80 of all LCDs).





### 3.3. LCDs in the Rhine

The main LCD in the SPM (n=5 in March; n=5 in September) of the Rhine is the $C_{32}$ 1,15-diol (2.7±1.2 ng.L$^{-1}$ in March; 4.6±2.5 ng.L$^{-1}$ in September) followed by the $C_{30}$ 1,15-diol (0.7±0.2 ng.L$^{-1}$ in March; 2.5±0.9 ng.L$^{-1}$ in September). The $C_{30}$

1,13-diol (0.5±0.2 ng.L$^{-1}$ in March; 0.6±0.2 ng.L$^{-1}$ in September) and $C_{28}$ 1,13-diol (0.3±0.1 ng.L$^{-1}$ in March; 1.2±0.3 ng.L$^{-1}$ in September) are also present, while the $C_{30}$ 1,14-diol (0.1±0 ng.L$^{-1}$ in March; 0.4±0.2 ng.L$^{-1}$ in September) and $C_{28}$ 1,14-diol (0.1±0 ng.L$^{-1}$ in March and 0.2±0.1 ng.L$^{-1}$ in September) are only minor compounds. The concentrations of the $C_{32}$ 1,15-diol was the highest at the sampling location in Karlsruhe in September with 9.1 ng.L$^{-1}$ (see Fig. 3b) and varies from 1.6 - 9.1 ng.L$^{-1}$ for all sites, while the $C_{30}$ 1,15-diol varies from 1.0 - 7.6 ng.L$^{-1}$. The LCD concentration is significantly higher in September

than in March (sum of all diols is 9.4±3.8 ng.L$^{-1}$ and 4.3±1.5 ng.L$^{-1}$, respectively, p<0.001). The BIT index varies from 0.64 to 0.93 (Fig. 3b), and is higher in September than in March (average 0.92±0.01 and 0.79±0.02, respectively). In March there is an increase of the BIT index downstream (0.64 in Karlsruhe to 0.86 in Kleve) but it remains constant in September. The chlorophyll concentrations vary from 1 to 6 µg.L$^{-1}$ with highest amount almost always in September (4±2 and 3±1 µg.L$^{-1}$ in September and March, respectively) indicating a small seasonal difference.

We sequenced the 18S rRNA gene from the Rhine SPM using universal eukaryote primers. Overall the libraries were dominated by reads affiliated to Opisthokonta (31 %), Stramenopiles (28 %), Hacrobia (24 %) and Alveolata (10 %). All the LCD-producing phytoplankton known to date (Eustigmatophyceae, *Proboscia spp.*, and *Apedinella radians*) are affiliated to the Stramenopile supergroup, and the Stramenopiles found here mostly include diatoms and Chrysophyceae. However the diatom OTUs most closely related to *Proboscia* belong to the genera *Melosira*, *Aulacoseira* , and *Actinocyclus* which have

never been reported to contain LCDs. The presence of LCDs within Chrysophyceae has also never been determined.
One eustigmatophyceaen OTU (denovo161 *Monodus guttula*, Supplement 2) represented by 2 reads (1 in Koblenz and 1 in Kleve in September) and 5 18S rRNA gene reads (1 in Cologne in March, 3 in Mainz, 1 in Koblenz and 2 in Kleve in September) associated with two OTUs from Pedinellales (denovo18 unidentified Pedinellales and denovo338 *Pseudopedinella sp.*, Supplement 2) were found in the Rhine SPM. The concentration of 18S rRNA genes varied between 2.7±0.1x10$^7$ and

1.0±0.1x10$^8$ copies.L-1 in March and 1.6±0.1x10$^7$ and 4.7±0.7x10$^7$copies.L-1 in September (Fig. 5c). Spearman rank correlation analyses performed using QIIME indicate that none of the OTU found here exhibit significant correlation with LCDs (data not shown).

### 4. Discussion

**4.1. Where are LCDs produced in rivers?**

For the different river systems a link between water conditions and LCD concentrations can often be observed. In particular, for the Godavari SPM there is a higher concentration of LCDs in the SPM from the wet season compared to that of the dry





season (1900±1000 ng.L⁻¹ and 570±260 ng.L⁻¹, respectively). The Godavari sediments collected during the dry season have higher concentration in LCDs (t-test, p <0.05) than the riverbed sediments from the wet season, opposite to what is observed in the SPM. During the wet season, the Godavari is more turbid and has a higher flow velocity. This high turbidity of the river water (Balakrishna and Probst, 2005; Syvitsky and Saito, 2007) may reduce LCD production as this limits light availability

and, therefore, algal productivity. However, the higher concentration of LCDs observed in the SPM during the wet season indicates that this explanation is likely not valid. Alternatively, the low concentration of LCDs in the wet season sediments could be due to the high flow velocity of the river water that prevents the LCDs formed in the rivers to be deposited in the river bed sediments.

During the wet season, the $C_{32}$ 1,15-diol is present in significantly (t-test, p<0.001) higher quantities in the sediments of the

Dowleswaram reservoir (500±80 ng.g⁻¹ sed) compared to other parts of the river system indicating that either the $C_{32}$ 1,15-diol production is enhanced in the reservoir or that the $C_{32}$ 1,15-diol is transported from upstream and accumulates within the reservoir. In the reservoir, both SPM and sediments have a higher organic carbon content compared to the rest of the river, whereas the suspended particle load is only slightly lower (Usman et al., 2018) which could be explained by contribution of primary produced organic carbon. This suggests that increased production of the $C_{32}$ 1,15-diol is facilitated by the calm

stagnant conditions in the reservoir. Similarly, in the Danube system the highest concentration of the $C_{32}$ 1,15-diol is also found in a calm and stagnant water area: the Iron Gates reservoir (5500 ng.g⁻¹ sed against 3200±1100 ng.g⁻¹ sed in average for the rest of the catchment, Fig. 2b).

Pradhan et al. (2014) reported a major input (i.e. 40-45%) of OM from freshwater algae in sediments from the Upper Godavari, based on the C/N ratio and $\delta^{13}C$ values of the sedimentary organic carbon, suggesting optimal conditions for aquatic production

in impoundments in this driest part of the river basin. Indeed, significantly (t-test, p<0.05) higher concentrations of the $C_{32}$ 1,15-diol are found in the Upper Godavari sediments collected during the dry season compared to the rest of the catchment, indicating that low flow, calm and stagnant water conditions are optimal for $C_{32}$ 1,15-diol production.

Collectively, our results suggests that LCDs and especially the $C_{32}$ 1,15-diol are preferentially produced in relatively calm and stagnant areas of river systems.


### 4.2. River versus marine LCDs

During the dry season, the SPM in the delta of the Godavari exhibits a LCD distribution significantly (t-test, p<0.05) deviating from the general distribution in river SPM and sediments, i.e. the LCDs are dominated by the $C_{30}$ 1,15-diol (fractional abundance of 0.74±0.14 for the delta *vs.* 0.47±0.16 for the rest of the river in the dry season) rather than the $C_{32}$ 1,15-diol. This

dominance of the $C_{30}$ 1,15-diol is different from LCD distributions usually found in rivers (De Bar et al., 2016; Lattaud et al., 2017a, this study), but similar to that observed in tropical marine sediments (Rampen et al., 2014a; Lattaud et al., 2017a). This suggests a marine influence on the LCDs in the delta during the dry season. Indeed, the electrical conductivity of the delta





river water in this season is typical for brackish water (Gupta et al., 1997; Sarma et al., 2009, 2010). The electrical conductivity decreases land inwards, indicating that the influence of marine waters is substentially reduced upstream.

The LCDs distribution in the Black Sea sediments are also dominated by the $C_{30}$ 1,15-diol (fractional abundance >0.9, Fig. 2c), whereas the $C_{32}$ 1,15-diol is the most abundant diol in the sediments of the Danube. The fractional abundance of the $C_{32}$

1,15-diol decreases with increasing distance from the river mouth as do the values for the BIT index (Fig. 2c). This decrease in $C_{32}$ 1,15-diol abundance is similar to that observed in the delta of the Godavari River during the dry season, but can now be followed along a much larger gradient; the Black Sea sediment has a clear marine signal (BIT=0.06±0.03, n=12, this study and Kusch et al., 2016) with a dominant $C_{30}$ 1,15-diol, whereas the Danube River is dominated by the $C_{32}$ 1,15-diol and has an average BIT index value of 0.91±0.04 (n=43, Freymond et al., 2017).

To visualize the differences between marine and river LCD distributions, a ternary plot was generated with the poles representing the different fractional abundances of the LCDs: $C_{30}$ 1,15-diol, $C_{32}$ 1,15-diol and the sum of $C_{30}$ 1,13 and $C_{28}$ 1,13-diols (Fig. 4a). The 1,14-diols were excluded from this plot as they likely have a different biological source (Sinninghe Damsté et al., 2003; Rampen et al., 20011, 2014b). Data from the river SPM from this study have been included as well as river SPM and sediments from Lattaud et al. (2017a) and lake sediments from Rampen et al. (2014b). For the marine dataset,

the marine sediments from Lattaud et al. (2017b), Rampen et al. (2012) and De Bar et al. (2016) were used. This ternary diagram shows that river SPM and sediments, as well as the lake sediments contain a higher proportion of $C_{32}$ 1,15-diol than open marine surface sediments, where their fractional abundance nearly always is <10%. This major difference in the distribution of LCDs in marine and freshwater environments suggests that LCDs are likely produced by different organism in freshwater and marine systems (cf. Lattaud et al., 2017a). This difference is useful to differentiate river influenced sediments

and marine sediment (cf. Lattaud et al., 2017b).

### 4.3. Who is producing LCDs in river systems?

### 4.3.1. Comparison with culture data

In all of the three river systems investigated here the $C_{32}$ 1,15-diol is the major diol (average fractional abundance of 0.47±0.17

for Danube, Rhine, and Godavari), followed by the $C_{30}$ 1,15-diol (0.31±0.21) (Fig. 1-3). To constrain potential biological producers of the LCDs, the LCD distributions in river SPM from this study and those of Lattaud et al. (2017a) and De Bar et al. (2016) were plotted in another ternary diagram (Fig. 4b), along with the LCD distribution of cultured eustigmatophyte algae (data from Rampen et al., 2014b). This diagram uses the fractional abundances of the $C_{30}$ 1,15-diol, $C_{32}$ 1,15-diol and the sum of the $C_{30}$ 1,13 and $C_{28}$ 1,13-diols. SPM from delta regions with a clear marine contribution were excluded, i.e. SPM with low

BIT values (BIT<0.3).

Most of the river LCD distributions are similar to the LCD distribution of *Goniochloris sculpta* from the Goniochloridaceae family, especially for the Rhine (this study), Danube (this study) and Tagus (data from De Bar et al., 2016). However, LCD





distributions in rivers from a tropical region such as the Godavari and Amazon do not plot close to this species (Rampen et al., 2014b). This observation may point at a role of temperature in the distribution of LCDs, or at different producers in tropical freshwater systems. However, there is no significant correlation between the fractional abundance (excluding 1,14-diols) of the $C_{32}$ 1,15-diol, or other diols, in rivers and mean annual air temperature of the river catchment ($r^2 = 0.002$, p = 0.6). This is

in contrast to the study of Rampen et al. (2014b), where a positive relation between the growth temperature of cultures of eustigmatophyte families and the fractional abundance of $C_{32}$ 1,15-diol was observed.

Villanueva et al. (2014) determined the diversity and abundance of specific eustigmatophyte algae using 18S rRNA gene sequences of the SPM at different water depths in Lake Challa, tropical East Africa. Villanueva et al. (2014) found 214 eustigmatophycean sequences affiliated to 5 distinct phylogenetic clades one of which affiliated to the Goniochloridaceae

family, and four novel groups, two of which closely related to the Monodopsidaceae and Eustigmatophyceae families. This suggests a role of novel uncultured Eustigmatophyceae in LCD production in riverine ecosystems. Furthermore, they quantified LCDs in monthly sediment trap material from the middle of the lake and observed that LCD distributions varied on a seasonal basis. The proportion of $C_{32}$ 1,15-diol was highest in February and June, while the $C_{30}$ 1,15-diol dominated in April, indicating separate blooms of different LCD producers. Our results suggest that there are potentially unknown

eustigmatophycean LCD producers in river systems or that there may be multiple LCD producers, depending on the season and the location.

### 4.3.2. 18S rRNA gene sequencing analysis of SPM in the Rhine

An alternative approach to comparing culture data to lipid distribution for identifying producers of LCDs is to analyze the

DNA composition of river water in which LCDs are detected (cf Villanueva et al., 2014). To characterize the producers of the 1,13 and 1,15-diols we sequenced V4 region of the 18S RNA gene on the SPM from the Rhine using 454 sequencing. This sequencing effort yielded ca. 60 000 reads for the pooled samples (see Supplement 2) but we only detected 3 OTUs and 9 reads associated with potential LCD producers. Indeed, the near absence of eustigmatophyte reads in the Rhine SPM (Fig. 5a) suggests that they are not the major producers of LCDs. Also dichtyochophytes were not detected in all the SPM in contrast

to LCDs. There is no correlation between the OTUs found in the Rhine River and the concentration of 1,13- and 1,15-diols. To pinpoint the producers of the 1,14-diols, we investigated the diatom distribution in the Rhine water. *Proboscia sp.*, the only known diatom producing 1,14-diols (Sinninghe Damsté et al., 2003; Rampen et al., 2014b), is a marine diatom (Moita et al., 2003; Lassiter et al., 2006; Takahashi et al., 2011) and, consistently, was not detected in our libraries. Other genera from the same group as *Proboscia* (radial centric diatoms) were found at all sites in March, and are also found in Karlsruhe and Mainz

in September (Fig 5b.). To establish whether these diatoms represent a potential source of 1,14-diols, we estimated their abundance by quantifying the concentration of total 18S rRNA gene copies multiplied with the percentage of each OTU of the total reads. However, there is no correlation between the concentration of 1,14-diols and the number of gene copies per liter of radial centric diatoms ($r^2 = 0.08$; p value = 0.4).





### 4.3.3. Are LCDs coming from dead OM or *in situ* living organisms?

Interestingly, there is no significant correlation ($r^2=0.2$, $p=0.2$) between the concentration of chlorophyll a (Fig. 5d) and the concentration of total LCDs, 1,14-diols ,or 1,13- + 1,15-diols (Fig. 5a, b), i.e. there is no apparent link between primary production and LCD production. The lack of correlation between LCDs and both OTUs and Chl-a suggests that the LCDs in the Rhine are either not produced *in situ* or are derived from unknown organisms (Villanueva et al., 2014, Balzano et al., 2018). To distinguish if the LCDs are coming from dead organic matter we performed an incubation experiment using $^{13}$C-labelled bicarbonate. After 52 h, $^{13}$C incorporation was detected in lipids such as β-sitosterol (+120‰) indicating uptake by phytoplankton. However, at the same time, we did not detect any incorporation of $^{13}$C in LCDs, suggesting that the incubation time may be too short for LCD producers to take up the $^{13}$C or that the LCDs are not synthetized *in situ*. If LCD producers are photosynthetic eukaryotes as indicated by culture studies (Volkman et al., 1999; Rampen et al., 2007, 2014b) then the incubation time used in the experiment should be sufficient for them to take up the $^{13}$C-bicarbonate dissolved in the water, suggesting that the LCDs are likely not synthesized in any of the sampling locations of the Rhine.

The absence of *in situ* LCD production could be due to the high flow velocity at these sampling sites. As also observed in the Godavari and Danube, the high velocity flow areas show a lower abundance of LCDs, while low flow areas show a higher abundance of LCDs. It is likely that, in the Rhine, the LCDs would be produced in more stagnant waters like in lakes, or dead river branches, and that they would be more abundant in these areas. Thus, LCDs which are likely degraded more slowly than DNA, reflect a fossil signal while the DNA reflects an *in situ* signal. Similarly, Villanuaneva et al. (2014) showed that while LCD were abundantly present in the surface water of Lake Challa, the DNA of eustigmatophyte could not be detected.

## 5. Conclusion

We studied three river systems to determine where LCDs are produced in rivers, if their distribution is different from that of marine LCDs, and to constrain their producers. Confirming previous results, riverine LCDs show a striking difference from marine LCDs as they have a high fractional abundance of the $C_{32}$ 1,15-diol (>50%) while marine LCDs have generally more of the $C_{30}$ 1,15-diol (>50%). The $C_{32}$ 1,15-diol is more abundant in calm stagnant waters than in fast flowing parts of the rivers, indicating that they are likely produced in calmer water. Comparison of LCD distributions of Eustigmatophyceae cultures with those in rivers indicate that *Goniochloris* species might be an important 1,13 and 1,15-diols producer in some river systems. 18S rRNA gene analysis of one of these rivers, the Rhine, did, however, not lead to any identification of this species, nor did a labelling study using bicarbonate lead to labelling of long chain diols. This might indicate that LCDs in fast flowing parts of rivers are not coming from *in situ* living plankton but from stagnant waters of these river systems such as lakes or side ponds.

### Competing interests.

The authors declare that they have no conflict of interest.





**Acknowledgements**

We thank Gabriella Weiss, Anchelique Mets, Kirsten Kooijman and Jort Ossebaar for analytical help. Heike Robakowski and the LUBW Landesanstalt für Umwelt, Messungen und Naturschutz Baden-Württemberg, Dr. Peter Diehl and the LUWG, Dr.

Helmut Fisher and the Landesamt für Umwelt Rheinland-Pfalz, Dr. Georg Becker and the University of Cologne, Jochen Lacombe and the Internationale Messstation Bimmen-Lobith and Sophie Reiche for providing help with the sampling. Huub Zwart, Chris Martes (UU), Muhammed Usman (ETH), and Sayak Basu (IISER Kolkata) are thanked for help in the field during Godavari fieldwork, we thank the captain and crew of the R/V Pelagia of cruise 64PE408 and 64PE418.

This research has been funded by the European Research Council (ERC) under the European Union's Seventh Framework

Program (FP7/2007-2013) ERC grant agreement [339206] to S.S. The work was further supported by funding from the Netherlands Earth System Science Center (NESSC) through a gravitation grant (NWO 024.002.001) from the Dutch Ministry for Education, Culture and Science to JSSD and SS. The Godavari River project was funded by NWO-Veni grant #863.13.016 to F.P. The Danube River project ("CAPS-LOCK" and "CAPS-LOCK2"; #200021_140850) was founded by the Swiss National Science Foundation SNF by grant to T.E.

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




**Legend**

Figure 1: a. Location of the Godavari samples (Usman et al., 2018) with enhanced view of the delta region, b. concentrations of LCDs in sediments (collected during wet and dry seasons) and c. concentrations of LCDs in SPM (collected during wet and dry seasons).

Figure 2: a. Location of the Danube samples with enhanced view of the Danube river mouth samples, b. concentration of LCDs (this study) and BIT values (from Freymond et al., 2017) in the Danube sediments and c. fractional abundance of the LCDs in the Danube (average of the Upper, Middle and Lower Danube), Danube delta, Danube river mouth (stations P128 and P177) and Black Sea (12 stations).

Figure 3: a. Location of the Rhine samples and b. concentration of LCDs and BIT values (September and March) in the Rhine.

Figure 4: Ternary plot ($C_{28}$ 1,13; $C_{30}$ 1,13 and $C_{30}$ 1,15; $C_{32}$ 1,15-diols) of a. marine sediments (from Rampen et al., 2012, De Bar et al., 2016, Lattaud et al., 2017b), river sediments (this study, Lattaud et al., 2017a), lake sediment (from Rampen et al., 2014b) and river SPM (this study, Lattaud et al., 2017a) and b. cultivated algae (Rampen et al., 2014b) and river SPM (this study, Lattaud et al., 2017a).

Figure 5: Results of the 18S rRNA analysis of the Rhine water, a. gene copy per liter of eustigmatophytes and dictyochophytes and concentration of 1,13 and 1,15-diols as well as concentration of the $C_{32}$ 1,15-diol, b. gene copy number per liter of radial centric diatoms and concentration of 1,14-diols, c. total gene copy per liter and d. chlorophyll a concentration.





**Figure 1**







**Figure 2**





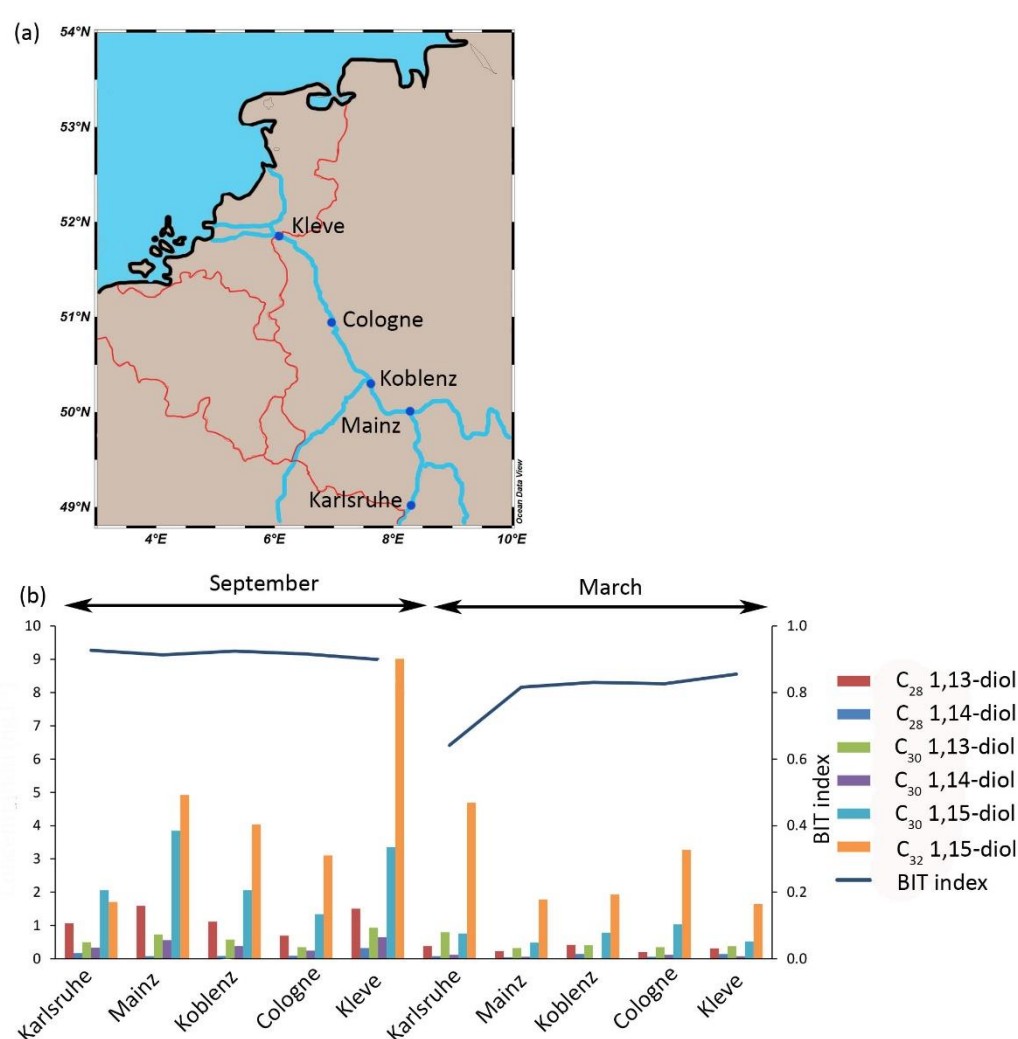

Figure 3



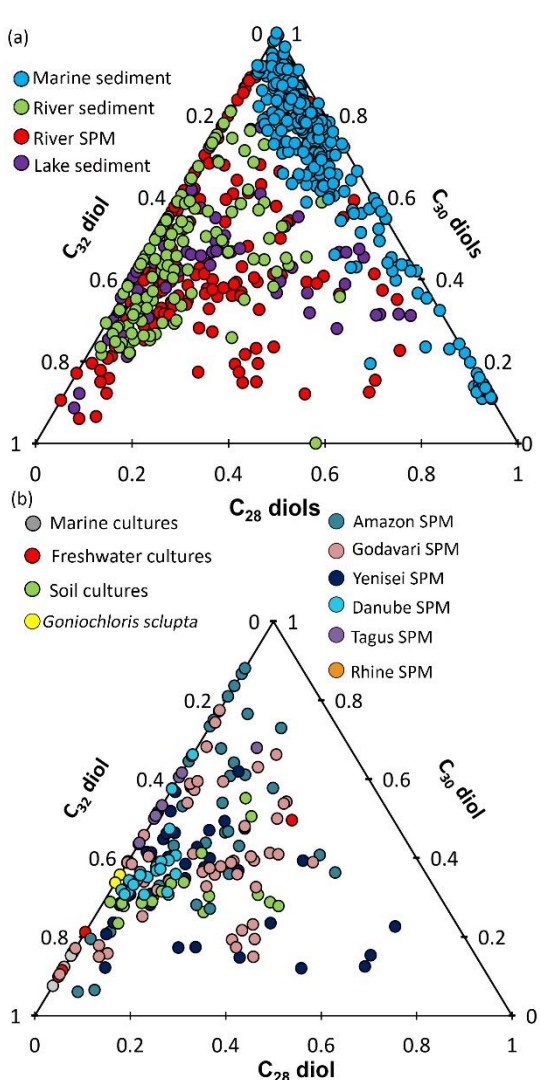

**Figure 4**



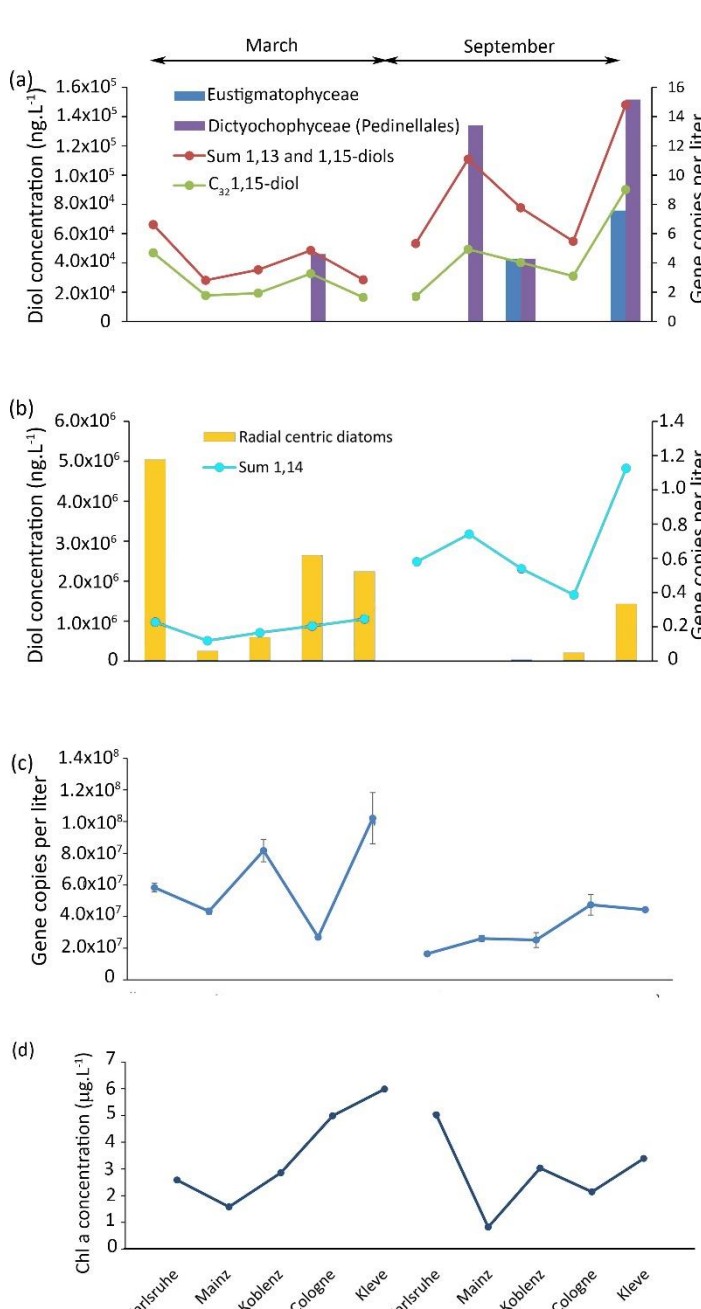

**Figure 5**