# Peer review of "Long-chain diols in rivers: Distribution and potential biological sources"

_Biogeosciences, 2018_

## Referee Comment (RC1) · Anonymous Referee #1 · 4 May 2018

**"Long-chain diols in rivers: distribution and potential biological sources"** by Julie Lattaud, et al

**General comments**:

This article describes the compositions of long-chain diols (LCDs) in the SPM of three river systems, and the effect of river flow and a seasonal change of LCDs. The main point is to attempt to elaborate the sources of LCDs in the river systems. Although these compounds are not so novel, the result in the article is suggestive for the bio-geochemical application of these compounds in the future.

The critical problem is the conclusion that LCDs in fast flowing parts of rivers are not coming from *in situ* living plankton but from stagnant waters of these river systems such as lakes or side ponds. The authors should sample SPM and surface sediments in these lakes or side ponds.

**Specific comments:**

1. Page 11, Lines: 5−9: The authors could confirm it by sampling the SPM at different depth in the water column.

2. The authors analyzed the GDGTs. Except the BIT indices, what other information could get from the GDGTs?

3. What the relationship between the temperature, precipitation and LCDs ?

---

## Referee Comment (RC2) · J. Plancq (Referee) · 9 May 2018

**Review on bg-2018-116 "Long-chain diols in rivers: distribution and potential biological sources" by Lattaud et al.**

This manuscript investigates the distribution of long-chain diols (LCDs) in suspended particulate matter (SPM) of three river systems in relation with season, precipitation, temperature, and source catchments. Confirming previous results, riverine LCDs show a striking difference from marine LCDs, with the dominance of the $C_{32}$ 1,15-diol in all investigated river systems. Higher concentrations of the $C_{32}$ 1,15-diol are also observed in stagnant water and during seasonal river low stands. 18S rRNA gene sequencing of SPM from the Rhine and isotope incubation of the river water suggest that the LCD-producers in rivers predominantly reside in lakes or side ponds that are part of the river system.

**General comments:**

This paper is a valuable contribution to the understanding of LCD genesis and relationships with environmental variables in fluvial environments. Indeed, there is currently only a limited body of literature on LCD source organisms in general and even more so in fluvial environments. The authors address relevant questions and provide novel concepts and data. The writing style is clear and precise, especially in the methods section, which provides sufficient information to replicate the results. The experiments were conducted with rigour and on substantial numbers of samples per river. The interpretations are most of the time supported by the data. This manuscript is thus suitable for Biogeosciences.

However, the current manuscript can be improved before publication. The authors discuss their finding in context of seasonality but sampling was only done during two separate months per year, which makes difficult to make inferences about the seasonality. It would be also nice to show the non-correlation between temperature and LCD fractional abundances of different isomers in the data section and not as a minor mentioning within the discussion. This aspect could have been discussed a bit more in the context of stronger temperature gradients between lakes and air temperature versus rivers and air temperature. Those two points become even more relevant since the abstract suggests a focus on LCD in relation to season, precipitation and temperature.

It is also unclear why GDGTs (BIT index) and Chlorophyll a are relevant for the outlined questions or are helpful in understanding spatial distributions of LCD and their source organisms. Although Chlorophyll a is sometimes used as indicator of primary productivity, it has been shown that there are more suitable parameters (Lyngsgaard et al., 2017). The result description could also be a bit more concise and part of the results shown in tables. Why did the authors not take SPM samples integrated over a greater part of the water column? Villanueva et al. (2014) showed maximum LCD concentration a few meters below the surface, and even though this is based on a lake, stagnant parts of the river systems could have a similar vertical distribution.

**Specific comments:**

| Page | Line | Comment |
|---|---|---|
| 1 | 20, 26 | Not only SPM but also sediment samples were investigated; "…in relation with season, precipitation, temperature, and source catchments" may be misunderstood as you making statements in all three rivers about seasonality even though SPMs were only sampled once during spring and once during autumn per location. Stating in the abstract that the relationship between LCD and temperature/precipitation was investigated and then only mentioning no correlation in the discussion may be perceived as misleading. |
| 2 | 6 | It would be clearer if you write that those culture experiments have been made on marine, lacustrine, soil and in snow living species. |
| 2 | 7 | Please be more specific. I think what you mean is that LCD signature of marine core top samples differ significantly from those of marine and lacustrine eustigmatophyte algae cultures. It would be good to state here that in marine versus lacustrine environments the $C_{32}$ 1,15 is less abundant that the $C_{30}$ 1,15-diol. |
| 2 | 19 | See comment page 2, line 6 |
| 3 | 16 | SPM of which water depth interval? |
| 5 | 1-2 | Why were the filters of the Rhine and the incubation experiment base hydrolyzed (Page 5, Lines 1-2) and not the other samples? It would be worth explaining the reason why in the methods. |
| 5 | 19-20 | It is said that the $C_{22}$ 7,16-diol was used as internal standard, while in Page 6, Lines 23-24, the $C_{22}$ 5,17-diol is indicated as internal standard. Is it a typo? |
| 7 | 28 | Please use the names of the primers provided by Stoeck et al. (2010). Why has this primer been used when it only yielded low quality V4 reads in the original paper? Organism-specific abundances may be biased by the quality of primer annealing to the template. It should be mentioned that denovo sequencing has been done. These constraints should be discussed.
 More specific primers could probably be used in future work instead of the universal eukaryotic primer. Since so far LCD producers all belong to the heterokonts, a primer specific to this algae group adapted to NGS would potentially yield better results (Coolen et al., 2004, Bittner et al., 2012). |
| 9 | 8 | Why writing the unit with a dot ("ng.L-1")? More commonly written without the dot. |
| 9 | 9 | Since figure 1c is referred to before 1b, I would change the numbering of figures to match the order of their mentioning. |
| 9 | 26-28 | The LCDs from Black Sea sediments have been quantified but only the fractional abundances are discussed in the text and shown in Figure 2c. All the other data (from the Basin, Reservoir and Delta) are absolute quantifications. Why? |
| 10 | 15 | Why not providing a cumulative column diagram for the different groups found in the DNA analysis? |
| 12 | 11, 28 | Discrepancies between text and ternary plot labels. Plot suggests all C28, C30 and C32 diols but in the text it is written as C30 1,15 diols, C32, 1,15 diols… |
| 13 | 11 | Why does the DNA work in lake Challa by Villanueva et al. (2014) suggest a role of novel uncultivated eustigmatophytes in LCD production in riverine ecosystems? |
| 14 | 2 | It is known that different algae have different chlorophyll signatures and chlorophyll a is very common. It was therefore unlikely that a relationship could have been found. Additionally, chlorophyll a is not necessary the best indicator of primary productivity. |
| 14 | 7 | Did the authors also do incubation experiments on waters in dead arms? [13]C may have been unsuccessful because LCD may have been produced in situ during blooms and incubation experiments may have been done on post-blooming waters. Please include those aspects in discussion. |

| Notes on figures | | |
|---|---|---|
| | 1a | It would be good to extent the white frame of the overview map further so that the labels are all within white background. Since reading the actual elevations is irrelevant and within the work area also not changing, I would reduce the labels on the scale to 0 to 4500 m. |
| | 1b,c | Since fig b and c use the same x scale, it would be better to put them closer together. Please write in the caption what those error bars represent. Standard error? 95%CI? Variability? (Same for Figures 1 and 3). |
| | 2a | Scale as discussed in 1a. |
| | 2b,c | Why BIT index is shown far away from marine influence? I guess there are no error bars for the Reservoir samples because there was only one sample. This should be specified in the figure caption. |
| | 3a,b | Please write if the samples are from sediment or SPM. Why no error bars here? Why BIT index? |
| | 4a,b | This Figure could be improved by using different symbols, for example in 4b different symbols could be used to distinguish more easily the culture samples from the SPM samples. The description of Figure 4a in Page 12, Lines 11-12 is different from the figure caption. The same applies for Figure 4b (Page 12, |

| | | |
|---|---|---|
| | | Lines 28-29). Please clarify. As the 1,14-diols were excluded, it would be appropriate to specify on the Figure "$C_{30}$ 1,13+1,15-diols" instead of only "$C_{30}$ diols". |
| | 5a,b,c | Please harmonise axis labels concerning diol concentrations. Why do gene copies have sometimes error bars and diols concentrations never have them? Why is chlorophyll a concentration shown without error bars? Why does the LCD concentration sometimes have error bars and sometimes not? |

**References:**

Bittner, L., et al. (2012). "Diversity patterns of uncultured Haptophytes unravelled by pyrosequencing in Naples Bay." Molecular Ecology **22**(1): 87-101.

Coolen, M. J. L., et al. (2004). "Combined DNA and lipid analyses of sediments reveal changes in Holocene haptophyte and diatom populations in an Antarctic lake." Earth and Planetary Science Letters **223**(1–2): 225-239.

Lyngsgaard, M. M., et al. (2017). "How Well Does Chlorophyll Explain the Seasonal Variation in Phytoplankton Activity?" Estuaries and Coasts **40**(5): 1263-1275.

Stoeck, T., et al. (2010). "Multiple marker parallel tag environmental DNA sequencing reveals a highly complex eukaryotic community in marine anoxic water." Molecular Ecology **19**(s1): 21-31.

Villanueva, L., et al. (2014). "Potential biological sources of long chain alkyl diols in a lacustrine system." Organic Geochemistry **68**: 27-30.

Julien Plancq and Mike M. Zwick

---

## Author Comment (AC1) · 14 May 2018

Reply to the interactive comment of the reviewer 1 on "Long-chain diols in rivers: distribution and potential biological sources" We thank reviewer 1 for his/her helpful comments on our manuscript. Below follows our reply to the main comments. General comments "The critical problem is the conclusion that LCDs in fast flowing parts of rivers are not coming from in situ living plankton but from stagnant waters of these river systems such as lakes or side ponds. The authors should sample SPM and surface sediments in these lakes or side ponds." We agree with the reviewer that lakes and side ponds are the logical subject of a follow-up study. It is known from the literature that LCDs occur in most lakes investigated (e.g. Rampen et al., 2014). We are currently sampling several lakes that are part of a river system to prove the hypothesis

developed in our manuscript. The results of this sampling campaign will be reported in a future manuscript. Specific comments -1 "Page 11, lines 5-9, the authors should confirm it by sampling the SPM at different depth in the water column" A depth profile has been analyzed for the Godavari River. Three depths (0, 4, and 8 m below the river surface) were sampled close to the river mouth in both wet and dry seasons (W10 and G10, respectively, see supplement and as illustrated in figure 1). No consistent difference were found in diol distributions between the different depths. We will state this more explicitly in a revised manuscript. -2 "The authors analyzed the GDGTs, except the BIT index what other information could you get from the GDGTs?" The current manuscript is focused on the LCDs and their ability to trace fluvial input into the marine environment. Similarly, the BIT index can be used to trace soil and riverine OC transported to the marine environment by the same river, which makes it a straightforward parameter for comparison. The occurrence and distribution of GDGTs, as well as their ability to transfer environmental signals from the catchment to the marine sedimentary archive have been discussed extensively in Freymond et al. (2017) and Freymond et al. (2018) for the Danube River. The data for the Godavari River are subject of another, still ongoing study in this area on soil OC transport.

-3 "What is the relationship between temperature, precipitation and LCDs?" There is no relation between temperature and LCD as is written in lines 2-6, page 13. There might be an indirect link between precipitation and LCDs. For example, in the Godavari system the concentration of LCDs in the riverbed sediment is higher during the wet season than during the dry season, in contrast to the SPM, where LCD concentrations are higher during the dry season. This link will be more clearly stated in a revised manuscript.

References Rampen, S.W., Datema, M., Rodrigo-Gámiza, M., Schouten, S., Reichart, G.-J. and Sinninghe Damsté, J.S.: Sources and proxy potential of long chain alkyl diols in lacustrine environments, Geochimi. Cosmochimi. Acta, 144, 59-71, doi.org/10.1016/j.gca.2014.08.033, 2014. Freymond, C.V., Peterse, F., Fischer,

L.V., Filip, F., Giosan, L. and Eglinton, T.I.: Branched GDGT signals in fluvial sediments of the Danube River basin: Method comparison and longitudinal evolution, Org. Geochem., 103, 88-96, doi.org/10.1016/j.orggeochem.2016.11.002, 2017. Freymond, C.V., Kundig,N., Stark, C, Peterse, F., Buggle, B., Lupker, M., Platze, M., Blattmann, T.M., Filip, F., Giosan, L., and Eglinton, T.I.: Evolution of biomolecular loadings along a major river system, Geochim. Cosmochim. Act., 223, 389-404, 10.1016/j.gca.2017.12.010, 2018.

[Figure]

**Fig. 1.** Depth profile of the concentration of C32 1,15- and C30 1,15-diols in the Godavari River (location 10, close to the river mouth and 28, in the middle of the river).

---

## Author Comment (AC2) · 25 May 2018

Reply to the interactive comment of the reviewer 2 on "Long-chain diols in rivers: distribution and potential biological sources"

We thank Dr. Julien Plancq for his helpful comments on our manuscript. Below follows our reply to the main comments.

General comments

- *"However, the current manuscript can be improved before publication. The authors discuss their finding in context of seasonality but sampling was only done during two separate months per year, which makes difficult to make inferences about the seasonality. It would be also nice to show the non-correlation between temperature and LCD fractional abundances of different isomers in the data section and not as a minor mentioning within the discussion. This aspect could have been discussed a bit more in the context of stronger temperature gradients between lakes and air temperature versus rivers and air temperature. Those two points become even more relevant since the abstract suggests a focus on LCD in relation to season, precipitation and temperature."*

We agree with the reviewer that only two months of sampling does not reflect a whole year of primary production. However the climate of the Godavari River system is characterized by only two distinct seasons, i.e a wet and a dry season. The sampling months have been chosen to reflect these two seasons. Samples from the Rhine River have been collected in spring and autumn, and thus does not reflect all seasons of the area. This will be specified in a revised version of the manuscript. We will also extend the discussion on the non-existent link between LCDs distributions and temperature.

- *"It is also unclear why GDGTs (BIT index) and Chlorophyll a are relevant for the outlined questions or are helpful in understanding spatial distributions of LCD and their source organisms. Although Chlorophyll a is sometimes used as indicator of primary productivity, it has been shown that there are more suitable parameters (Lyngsgaard et al., 2017). The result description could also be a bit more concise and part of the results shown in tables. Why did the authors not take SPM samples integrated over a greater part of the water column? Villanueva et al. (2014) showed maximum LCD concentration a few meters below the surface, and even though this is based on a lake, stagnant parts of the river systems could have a similar vertical distribution."*

We have used the BIT index as an indicator for marine input, especially in the lower part of the river. We have used chlorophyll a as indicator for primary production, as the data was readily available. Indeed, Lynsgaard et al. (2017) suggest that chlorophyll a failed to reflect the summer primary productivity in the Baltic Sea and that phytoplankton carbon biomass concentration is a better proxy. Unfortunately, we did not fix the phytoplankton and did not perform any biovolume counts. Therefore, we cannot obtain these phytoplankton biomass concentration values and chose to use chlorophyll a, imperfect as it is, instead.

The SPM from the water column of the Godavari River were taken at the surface (0 m), at the middle of the water column (2 or 4 m) and as close as the bottom as possible (8 m) to cover as much as possible the water column. The non relation with depth will be more emphasized in the revised manuscript.

Specific comments-Page 1, line 20, 26 *"Not only SPM but also sediment samples were investigated; "…in relation with season, precipitation, temperature, and source catchments" may be misunderstood as you making statements in all three rivers about seasonality even though SPMs were only sampled once during spring and once during autumn per location. Stating in the abstract that the relationship between LCD*

*and temperature/precipitation was investigated and then only mentioning no correlation in the discussion may be perceived as misleading."*

For the Godavari River, the two months sampled are reflecting the two seasons existing in this area, i.e. the wet and the dry season. For the Rhine River we agree that sampling in spring and autumn does not reflect all seasons of the area, and this will be nuanced in a revised version of the manuscript. The Danube samples covers a large temperature gradient where we could have detected a temperature/diols relation. The relationship between temperature and LCDs distribution is only briefly mentioned as no correlation found but we will clarify and somewhat extend that discussion in the revised manuscript. The relationship between precipitation and LCDs distribution was especially investigated in the Godavari River system as the two months of sampling differ mainly in precipitation, and is discussed on page 9 lines 8-18 and pages 10-11 lines 31-32 and 1-3 of the initial manuscript.

-Page 2, line 6 *"It would be clearer if you write that those culture experiments have been made on marine, lacustrine, soil and in snow living species.*

We will add this to the text in a revised manuscript.

-Page 2, line 7 *"Please be more specific. I think what you mean is that LCD signature of marine core top samples differ significantly from those of marine and lacustrine eustigmatophyte algae cultures. It would be good to state here that in marine versus lacustrine environments the C32 1,15 is less abundant that the C30 1,15-diol."*

We will clarify the sentence and specify the difference between the LCD signature in marine and lacustrine sediments.

-Page 2, line 19 *"See comment page 2, line 6.*

We will specify that the cultivated isolates are derived from different environments.

-Page 3, line 16 *"SPM of which water depth interval?"*

We will add that most of the SPM were collected at the surface and that water column profiles were sampled at two locations in the wet season and one in the dry season.

-Page 5, lines 1-2 *"Why were the filters of the Rhine and the incubation experiment base hydrolyzed (Page 5, Lines 1-2) and not the other samples? It would be worth explaining the reason why in the methods."*

The samples for the Godavari and Danube systems have been primarily studied for GDGTs, and have been treated as such. The standard protocol for LCDs extraction includes a base hydrolysis step to allow enhance recovery. However, a recent study (Reiche et al., in preparation) showed that base hydrolysis does not yield much more LCDs than direct extraction. Hence we expect that there will be little difference in the results between the two work-up procedures.

-Page 5, lines 19-20 *"It is said that the C22 7,16-diol was used as internal standard, while in Page 6, Lines 23-24, the C22 5,17-diol is indicated as internal standard. Is it a typo?"*

This is a typo, it should be the $C_{22}$ 7,16-diol, and will be corrected.

*-Page 7, line 28 "Please use the names of the primers provided by Stoeck et al. (2010). Why has this primer been used when it only yielded low quality V4 reads in the original paper? Organism-specific abundances may be biased by the quality of primer annealing to the template. It should be mentioned that denovo sequencing has been done. These constraints should be discussed. More specific primers could probably be used in future work instead of the universal eukaryotic primer. Since so far LCD producers all belong to the heterokonts, a primer specific to this algae group adapted to NGS would potentially yield better results (Coolen et al., 2004, Bittner et al., 2012)."*

The names of the primers are TAReuk454FWD1 and TAReuk454REV3 (Stoeck et al., 2010). The original paper compares V4 sequencing and V9 sequencing. While V9 primers seem indeed to capture the highest level of diversity, the V4 primers do not underperform at all (see Fig. 4 on Stoeck paper). These primers (Tareuk) have been extensively used for many years after the original publication and they are considered among the best primers for targeting microbial eukaryotes in the environment. Furthermore, the V4 region is a longer fragment than the V9, this allows a higher resolution in discriminating species from the same genus or clades within the same species. More specific primers can be used, like in Villanueva et al. (2014) where they used specific eustigmatophyte primers. However, we wanted, in this study, to look at all potential diol producers that includes organisms outside of the heterokont group (like *Azolla filliculoides*, Jetter et al., 2009).

*-Page 9, line 8 "9 Why writing the unit with a dot ("ng.L-1")? More commonly written without the dot."*

We will remove the dot in the revised manuscript.

*-Page 9, line 9 "Since figure 1c is referred to before 1b, I would change the numbering of figures to match the order of their mentioning."*

We will change the numbering of the figures accordingly.

*-Page 9, lines 26-28 "The LCDs from Black Sea sediments have been quantified but only the fractional abundances are discussed in the text and shown in Figure 2c. All the other data (from the Basin, Reservoir and Delta) are absolute quantifications. Why?"*

The Black Sea sediments have only been used to identify the marine end member in the ternary plots which is based on fractional abundances. We do have the absolute concentrations of the LCDs in the Black Sea and will add them to the results section of the revised manuscript.

*-Page 10, line 15 "Why not providing a cumulative column diagram for the different groups found in the DNA analysis?*

We will add a cumulative diagram in the supplement with the DNA sequencing results.

*-Page 12, lines 11, 28 "Discrepancies between text and ternary plot labels. Plot suggests all C28, C30 and C32 diols but in the text it is written as C30 1,15 diols, C32, 1,15 diols…"*

The legend of figure 4 is correct, we will correct the text to indicate that the three axes are the $C_{32}$ 1,15-diol, the $C_{28}$ 1,13-diol and the $C_{30}$ 1,13 + 1,15-diols.

*-Page 13, line 11 "Why does the DNA work in lake Challa by Villanueva et al. (2014) suggest a role of novel uncultivated eustigmatophytes in LCD production in riverine ecosystems?"*

Sequences of novel uncultivated eustigmatophytes were found in lake Challa and their gene copies distribution fit with the LCD distribution, suggesting they may be a source for LCDs.

*-Page 14, line 2 "It is known that different algae have different chlorophyll signatures and chlorophyll a is very common. It was therefore unlikely that a relationship could have been found. Additionally, chlorophyll a is not necessary the best indicator of primary productivity."*

We agree with the comments, but unfortunately we do not have the total chlorophyll available.

*-Page 14, line 7 "Did the authors also do incubation experiments on waters in dead arms? 13C may have been unsuccessful because LCD may have been produced in situ during blooms and incubation experiments may have been done on post-blooming waters. Please include those aspects in discussion."*

We did not perform any incubations in dead arms of rivers. We agree with the reviewer that the incubation could have failed because of the timing of the experiment (post-bloom) rather than because of the absence of the producers although it should be noted that we did detect incorporation in other algal biomarkers. We will add this aspect in the discussion.

Note on Figures

*-1a "It would be good to extent the white frame of the overview map further so that the labels are all within white background. Since reading the actual elevations is irrelevant and within the work area also not changing, I would reduce the labels on the scale to 0 to 4500 m."*

We will make the white frame bigger and reduce the elevation scale.

*-1b,c "Since fig b and c use the same x scale, it would be better to put them closer together. Please write in the caption what those error bars represent. Standard error? 95%CI? Variability? (Same for Figures 1 and 3)."*

We will put the figures closer together to make the reading clearer. We will add to the figure legend that the error bars represent standard deviation.

*-2a "Scale as discussed in 1a."*

We will also extend the white frame of the overview map to make the label more visible.

*-2b,c "Why BIT index is shown far away from marine influence? I guess there are no error bars for the Reservoir samples because there was only one sample. This should be specified in the figure caption."*

The BIT values for the parts removed from the sea are only to show that the riverine end member is very different from the marine end member. We will specify in the figure caption that there is only one sample for the reservoir.

*-3a, b "Please write if the samples are from sediment or SPM. Why no error bars here? Why BIT index?"*

We will add that the samples are from SPM. The BIT index here allows us to compare with the Danube River to clearly show the difference between marine and riverine environments.

*-4a, b "This Figure could be improved by using different symbols, for example in 4b different symbols could be used to distinguish more easily the culture samples from the SPM samples. The description of Figure 4a in Page 12, Lines 11-12 is different from the figure caption. The same applies for Figure 4b (Page 12,Lines 28-29). Please clarify. As the 1,14-diols were excluded, it would be appropriate to specify on the Figure "C30 1,13+1,15-diols" instead of only "C30 diols".*

We will change the symbols for culture samples for figure 4b. The figure caption is right, the description should be $C_{32}$ 1,15-diol; $C_{30}$ 1,13 + 1,15-diols and $C_{28}$ 1,13-diol. We will correct this in the text, and make a more detailed caption.

*-5a, b, c "Please harmonise axis labels concerning diol concentrations. Why do gene copies have sometimes error bars and diols concentrations never have them? Why is chlorophyll a concentration shown without error bars? Why does the LCD concentration sometimes have error bars and sometimes not?"*

Diol concentrations are only reflecting one point and one measurement, in contrast to the gene copies which are from triplicate analysis. Only one replica has been done to the chlorophyll concentration measurements, which is why there is no error bar. LCDs concentrations have error bars when they represent the average of a river section (for the Godavari and Danube systems). The LCD concentrations for the Rhine reflect only one site. We will specify this in the figure caption.

---

## Author Response (AR2)

Dear editor,

Please find resubmitted the manuscript entitled "Long-chain diols in rivers: distribution and potential biological sources" (bg-2018-116). We thank you, and the reviewers, for your positive comments and we have used these to revise and improve our manuscript. The reviewers and you requested minor changes and we took these into account. Below we have listed our responses (in bold) to the reviewers concerns and suggested changes.

We hope that with this revision we have addressed all issues and that the revised manuscript is suitable for publication in BG.

On behalf of the co-authors,

Sincerely,

Julie Lattaud

We are pleased to inform you that both reviewers suggested that the manuscript should be published subjected to minor revisions. Please respond to the reviewers' comments and submit a final version of your manuscript for potential publication in BG.

**Dear editor, we thank you for the positive feedback and addressed the reviewers' comments below and in the revised manuscript.**

Reviewer 1

We thank the anonymous reviewer 1 for her/his helpful comments on our manuscript. Below follows our reply to the main comments and, where applicable, how we changed the manuscript.

General comments

> *"The critical problem is the conclusion that LCDs in fast flowing parts of rivers are not coming from in situ living plankton but from stagnant waters of these river systems such as lakes or side ponds. The authors should sample SPM and surface sediments in these lakes or side ponds."*

> **We agree with the reviewer that lakes and side ponds are the logical subject of a follow-up study. It is known from the literature that LCDs occur in most lakes investigated (e.g. Rampen et al., 2014). We are currently sampling several lakes that are part of a river system to prove the hypothesis developed in our manuscript. The results of this sampling campaign will be reported in a future manuscript.**

Specific comments

> -1 "*Page 11, lines 5-9, the authors should confirm it by sampling the SPM at different depth in the water column*"
> **A depth profile has been analyzed for the Godavari River. Three depths (0, 4, and 8 m below the river surface) were sampled close to the river mouth in both wet and dry seasons (W10 and G10, respectively, see supplement and as illustrated in Figure 1). No consistent difference was found in diol distributions between the different depths. We added on page 3 line 18 "The SPM was sampled at the surface of the river except for two depth profiles (0, 4 and 8 m deep) which were sampled during the wet season and during the dry season, respectively".**

> -2 *"The authors analyzed the GDGTs, except the BIT index what other information could you get from the GDGTs?"*
> **The current manuscript is focused on the LCDs and their ability to trace fluvial input into the marine environment. Similarly, the BIT index can be used to trace soil and riverine OC transported to the marine environment by the same river, which makes it a straightforward parameter for comparison. The occurrence and distribution of GDGTs, as well as their ability to transfer environmental signals from the catchment to the marine sedimentary archive have been discussed extensively in Freymond et al. (2017) and Freymond et al. (2018) for the Danube River. The data for the Godavari River are subject of another, still ongoing study in this area on soil OC transport.**

> -3 *"What is the relationship between temperature, precipitation and LCDs?"*
> **There is no relation between temperature and LCD as is written in lines 2-6, page 13. There might be an indirect link between precipitation and LCDs but it is not clear from our data. For example, in the Godavari system the concentration of LCDs in the riverbed sediment is lower during the wet season than during the dry season, in contrast to the SPM, where LCD concentrations are lower during the dry season.**

Reviewer 2

**We thank Dr. Plancq for his helpful comments on our manuscript. Below follows our reply to the main comments and, where applicable, how we changed the manuscript.**

General comments

> *-"However, the current manuscript can be improved before publication. The authors discuss their finding in context of seasonality but sampling was only done during two separate months per year, which makes difficult to make inferences about the seasonality. It would be also nice to show the non-correlation between temperature and LCD fractional abundances of different isomers in the data section and not as a minor mentioning within the discussion. This aspect could have been discussed a bit more in the context of stronger temperature gradients between lakes and air temperature versus rivers and air temperature. Those two points become even more relevant since the abstract suggests a focus on LCD in relation to season, precipitation and temperature."*

**We agree with the reviewer that only two months of sampling does not reflect a whole year of primary production. However, the climate of the Godavari River system is characterized by only two distinct seasons, i.e a wet and a dry season. The sampling months have been chosen to reflect these two seasons. Samples from the Rhine River have been collected in spring and autumn, and thus does not reflect all seasons of the area. This is specified in the revised version of the manuscript (page 1, line 20, we removed the mention of seasonality). We added page 13 lines 10-11 "Potentially, this difference could be due to the fact that we are using mean annual air temperature and not in situ river temperatures".**

> *-"It is also unclear why GDGTs (BIT index) and Chlorophyll a are relevant for the outlined questions or are helpful in understanding spatial distributions of LCD and their source organisms. Although Chlorophyll a is sometimes used as indicator of primary productivity, it has been shown that there are more suitable parameters (Lyngsgaard et al., 2017). The result description could also be a bit more concise and part of the results shown in tables. Why did the authors not take SPM samples integrated over a greater part of the water column? Villanueva et al. (2014) showed maximum LCD concentration a few meters below the surface, and even though this is based on a lake, stagnant parts of the river systems could have a similar vertical distribution."*

**We have used the BIT index as an indicator for marine input in river systems, especially in the lower part of the river. We have used chlorophyll a as indicator for primary production, as the data was readily available. Indeed, Lynsgaard et al. (2017) suggest that chlorophyll a failed to reflect the summer primary productivity in the Baltic Sea and that phytoplankton carbon biomass concentration is a better proxy. Unfortunately, we did not fix the phytoplankton and did not perform any biovolume counts. Therefore, we cannot obtain these phytoplankton biomass concentration values and chose to use chlorophyll a instead.**

**The SPM from the water column were taken at the surface (0 m), at the middle of the water column (2 or 4 m) and as close as the bottom as possible (8 m) to cover as much as possible the water column. We added these details on page 3 lines 18-19 "The SPM was sampled at the surface of the river except for two depth profiles (0, 4 and 8 m deep) which were sampled during the wet season and during the dry season, respectively".**

Specific comments

> -Page 1, line 20, 26 *"Not only SPM but also sediment samples were investigated; "...in relation with season, precipitation, temperature, and source catchments" may be misunderstood as you making statements in all three rivers about seasonality even though SPMs were only sampled once*

*during spring and once during autumn per location. Stating in the abstract that the relationship between LCD and temperature/precipitation was investigated and then only mentioning no correlation in the discussion may be perceived as misleading."*

**For the Godavari River, the two months sampled are reflecting the only two seasons existing in this area, i.e. the wet and the dry season. For the Rhine River we agree that sampling in spring and autumn does not reflect all seasons of the area, and this will be nuanced in a revised version of the manuscript. The Danube samples covers a large temperature gradient where we could have detected a temperature/diols relationship. The relationship between temperature and LCDs distribution is only briefly mentioned as no correlation found but we will clarify and somewhat extend that discussion in the revised manuscript (page 13 lines 10-11). The relationship between precipitation and LCDs distribution was especially investigated in the Godavari River system as the two months of sampling differ mainly in precipitation, and is discussed on page 9 lines 8-18 and pages 10-11 lines 31-32 and 1-3 of the initial manuscript.**

-Page 2, line 6 *"It would be clearer if you write that those culture experiments have been made on marine, lacustrine, soil and in snow living species."*
**We added page 2 lines 6-7 "(isolated from snow, soil, marine and freshwater environments)"**

-Page 2, line 7 *"Please be more specific. I think what you mean is that LCD signature of marine core top samples differ significantly from those of marine and lacustrine eustigmatophyte algae cultures. It would be good to state here that in marine versus lacustrine environments the C32 1,15 is less abundant that the C30 1,15-diol."*
**We clarified the sentence and specify the difference between the LCD signature in marine and lacustrine sediments page 2 lines 8-9 "i.e. cultures of eustigmatophytes produce mainly the $C_{32}$ 1,15-diol while in marine sediments the $C_{30}$ 1,15-diol is generally dominant".**

-*Page 2, line 19 "See comment page 2, line 6"*
**We think that our sentence has made it clear that the cultures studied were lacustrine only. Page 2, lines 19-20 "Rampen et al. (2014b) studied the LCD distribution of several freshwater eustigmatophyte cultures"**

-*Page 3, line 16 "SPM of which water depth interval?"*
**We added page 3 line 18 "The SPM was sampled at the surface of the river except for two depth profiles (0, 4 and 8 m deep) which were sampled during the wet season and during the dry season, respectively".**

-*Page 5, lines 1-2 "Why were the filters of the Rhine and the incubation experiment base hydrolyzed (Page 5, Lines 1-2) and not the other samples? It would be worth explaining the reason why in the methods."*
**The samples for the Godavari and Danube systems have been primarily studied for GDGTs and have been treated as such. The standard protocol for LCDs extraction includes a base hydrolysis step to allow enhance recovery. However, a recent study (Reiche et al., in preparation) showed that base hydrolysis does not yield much more LCDs than direct extraction. Hence there will be little difference in the results between the two work-up procedures.**

-*Page 5, lines 19-20 "It is said that the C22 7,16-diol was used as internal standard, while in Page 6, Lines 23-24, the C22 5,17-diol is indicated as internal standard. Is it a typo?"*
**This is a typo, it should be the $C_{22}$ 7,16-diol, it is corrected accordingly. Page 6 line 26 "$C_{22}$ 7,16-diol"**

*-Page 7, line 28 "Please use the names of the primers provided by Stoeck et al. (2010). Why has this primer been used when it only yielded low quality V4 reads in the original paper? Organism-specific abundances may be biased by the quality of primer annealing to the template. It should be mentioned that denovo sequencing has been done. These constraints should be discussed.*
*More specific primers could probably be used in future work instead of the universal eukaryotic primer. Since so far LCD producers all belong to the heterokonts, a primer specific to this algae group adapted to NGS would potentially yield better results (Coolen et al., 2004, Bittner et al., 2012)."*

**The names of the primers are TAReuk454FWD1 and TAReuk454REV3 (Stoeck et al., 2010), they have been added on page 7, lines 31 and 32. The original paper compares V4 sequencing and V9 sequencing. While V9 primers seem indeed to capture the highest level of diversity, the V4 primers do not underperform at all (see Fig. 4 on Stoeck paper). These primers (Tareuk) have been extensively used for many years after the original publication and they are considered among the best primers for targeting microbial eukaryotes in the environment. Furthermore, the V4 region is a longer fragment than the V9, this allows a higher resolution in discriminating species from the same genus or clades within the same species. More specific primers can be used, like in Villanueva et al. (2014) where they used specific eustigmatophyte primers. However, we wanted, in this study, to look at all potential diol producers that includes organisms outside of the heterokont group (like *Azolla filliculoides*, Jetter et al., 2009).**

*-Page 9, line 8 "9 Why writing the unit with a dot ("ng.L-1")? More commonly written without the dot."*
**We have removed all dots from units.**

*-Page 9, line 9 "Since figure 1c is referred to before 1b, I would change the numbering of figures to match the order of their mentioning."*
**We changed figure 1 accordingly.**

*-Page 9, lines 26-28 "The LCDs from Black Sea sediments have been quantified but only the fractional abundances are discussed in the text and shown in Figure 2c. All the other data (from the Basin, Reservoir and Delta) are absolute quantifications. Why?"*
**The Black Sea sediments have only been used to identify the marine end member in the ternary plots which is based on fractional abundances. We do have the absolute concentrations of the LCDs in the Black Sea, and they have been added on page 10 lines 1-2 "In the Black sea sediment the main LCD was the $C_{30}$ 1,15-diol ($6500\pm9000$ ngg$^{-1}$ sed) followed by the $C_{30}$ 1,14-diol ($1100\pm1600$ ngg$^{-1}$ sed)".**

*-Page 10, line 15 "Why not providing a cumulative column diagram for the different groups found in the DNA analysis?"*

**We added a cumulative diagram as supplement 3 with the DNA sequencing results.**

*-Page 12, lines 11, 28 "Discrepancies between text and ternary plot labels. Plot suggests all C28, C30 and C32 diols but in the text it is written as C30 1,15 diols, C32, 1,15 diols…"*

**The plots legend of figure 4 is correct, we corrected the text, page 12, lines 29-30 "This diagram uses the fractional abundances of the $C_{28}$ 1,13-diol, $C_{32}$ 1,15-diol and the sum of the $C_{30}$ 1,13 and $C_{30}$ 1,15-diols".**

*-Page 13, line 11 "Why does the DNA work in lake Challa by Villanueva et al. (2014) suggest a role of novel uncultivated eustigmatophytes in LCD production in riverine ecosystems?"*

**Sequences of novel uncultivated eustigmatophytes were found in Lake Challa and their gene copies distribution fit with the LCD distribution, suggesting they may be a source for LCDs.**

*-Page 14, line 2 "It is known that different algae have different chlorophyll signatures and chlorophyll a is very common. It was therefore unlikely that a relationship could have been found. Additionally, chlorophyll a is not necessary the best indicator of primary productivity."*

**We agree with the comments, unfortunately we do not have the total chlorophyll data available.**

*-Page 14, line 7 "Did the authors also do incubation experiments on waters in dead arms? 13C may have been unsuccessful because LCD may have been produced in situ during blooms and incubation experiments may have been done on post-blooming waters. Please include those aspects in discussion."*

**We did not perform any incubations in dead arms of rivers. We agree with the reviewer t that the incubation could have failed because of the timing of the experiment (post-bloom) rather than because of the absence of the producers although it should be noted that we did detect incorporation in other algal biomarkers. We added on page 14, lines 13-15 "suggesting that the incubation time may be too short for LCD producers to take up the $^{13}$C or that the LCDs are not synthesized *in situ* during the time of sampling".**

Note on Figures

*-1a "It would be good to extent the white frame of the overview map further so that the labels are all within white background. Since reading the actual elevations is irrelevant and within the work area also not changing, I would reduce the labels on the scale to 0 to 4500 m."*

**We made the white frame bigger and reduce the elevation scale. See figure 1 page 20.**

*-1b,c "Since fig b and c use the same x scale, it would be better to put them closer together. Please write in the caption what those error bars represent. Standard error? 95%CI? Variability? (Same for Figures 1 and 3)."*

**We put the figures closer together to make the reading clearer. We added to the figure legend that the error bars represent the standard deviation. Page 19, line 4 and page 20, figure 1.**

*-2a "Scale as discussed in 1a."*
**We extended the white frame of the overview map to make the label more visible.**

*-2b,c "Why BIT index is shown far away from marine influence? I guess there are no error bars for the Reservoir samples because there was only one sample. This should be specified in the figure caption."*
**The BIT values for the part removed from the sea are only to show that the riverine end member is very different from the marine end member. We specified that there is only one sample for the reservoir, page 19 line 6 "(n=1 for the reservoir)".**

*-3a, b "Please write if the samples are from sediment or SPM. Why no error bars here? Why BIT index?"*
**We added that the samples are from SPM, page 19, line 9-10 "concentration of LCDs and BIT values (September and March) in the Rhine SPM.". The BIT index here allows us to compare**

**with the Danube River to clearly show the difference between marine and riverine environments.**

*-4a, b "This Figure could be improved by using different symbols, for example in 4b different symbols could be used to distinguish more easily the culture samples from the SPM samples. The description of Figure 4a in Page 12, Lines 11-12 is different from the figure caption. The same applies for Figure 4b (Page 12, Lines 28-29). Please clarify. As the 1,14-diols were excluded, it would be appropriate to specify on the Figure "C30 1,13+1,15-diols" instead of only "C30 diols""*
**We changed the symbols for cultures samples for figure 4b. The figure caption is correct, the description should be $C_{32}$ 1,15-diol; $C_{30}$ 1,13 + 1,15-diols and $C_{28}$ 1,13-diol. We therefore corrected it in the text (page 12, lines 29-30), and we made a more detailed caption.**

*-5a, b, c "Please harmonise axis labels concerning diol concentrations. Why do gene copies have sometimes error bars and diols concentrations never have them? Why is chlorophyll a concentration shown without error bars? Why does the LCD concentration sometimes have error bars and sometimes not?"*

**Diol concentrations are only reflecting one point and one measurement, in contrast to the gene copies which are from triplicate analysis. Only one replica has been done to measure the chlorophyll concentration, which is why there are no error bars. LCDs concentrations have error bars when they represent the average of a river section (for the Godavari and Danube systems). The LCD concentrations for the Rhine reflect only one site. We specify this in the figure caption, page 19 line 9 "n=1 per site".**

[revised manuscript text omitted]